# Non-Antigenic Modulation of Antigen Receptor (TCR) Cβ-FG Loop Modulates Signalling: Implications of External Factors Influencing T-Cell Responses

**DOI:** 10.3390/ijms24119334

**Published:** 2023-05-26

**Authors:** Nicholas Manolios, Son Pham, Guojiang Hou, Jonathan Du, Camelia Quek, David Hibbs

**Affiliations:** 1Faculty of Medicine and Health, The University of Sydney, Sydney, NSW 2006, Australia; 2Department of Rheumatology, Westmead Hospital, Sydney, NSW 2145, Australia; kevinhou501@hotmail.com; 3Sydney Medical School, Faculty of Medicine and Health, The University of Sydney, Sydney, NSW 2006, Australia; spha8630@alumni.sydney.edu.au (S.P.); camelia.quek@sydney.edu.au (C.Q.); 4The University of Sydney School of Pharmacy, Faculty of Medicine and Health, The University of Sydney, Sydney, NSW 2006, Australia; jonathan.james.sing.on.du@emory.edu (J.D.); david.hibbs@sydney.edu.au (D.H.)

**Keywords:** T-cell antigen receptor, computer modelling, in silico, short chain fatty acids, T-cell signalling, environmental factors

## Abstract

T-cell recognition of antigens is complex, leading to biochemical and cellular events that impart both specific and targeted immune responses. The end result is an array of cytokines that facilitate the direction and intensity of the immune reaction—such as T-cell proliferation, differentiation, macrophage activation, and B-cell isotype switching—all of which may be necessary and appropriate to eliminate the antigen and induce adaptive immunity. Using in silico docking to identify small molecules that putatively bind to the T-cell Cβ-FG loop, we have shown in vitro using an antigen presentation assay that T-cell signalling is altered. The idea of modulating T-cell signalling independently of antigens by directly targeting the FG loop is novel and warrants further study.

## 1. Introduction

The T-cell antigen receptor (TCR) is a multi-subunit complex composed of integral membrane proteins on the cell surface (Figure 1A) which collectively exhibit dual functions of antigen recognition and signal transduction [1]. The TCR-α/β chains engage in antigen recognition, while the CD3-e, γ, δ and ζ-ζ chains participate in signal transduction. The TCR plays a crucial role in human biology in maintaining immune homeostasis, and under certain pathological conditions or altered states of function, can lead to infections, cancer, inflammation, and autoimmunity [1]. X-ray crystallography of the TCR-β chain has defined a unique protrusion of the FG loop of the Cβ domain, linking its βF and βG strands to the co-accessory CD3-ε protein. This raises considerable interest at this site for protein–protein interactions involved with antigen recognition, signal transduction, and subsequent translational T-cell events (Figure 1B) [2]. The FG loop is conserved in the TCR-αβ complex across all mammalian species [3]. Removal of the conserved region of the FG loop or deletion of the TCR-Cβ-FG loop results in the formation of T-cell mutants [4,5] impairs the selection of T-cell subsets, and causes the cells to exit the phenotypic development of T-cell differentiation [6,7]. Furthermore, there is a mounting body of evidence highlighting the involvement of the TCR-Cβ-FG loop in T-cell signalling [3,5,6,8,9,10], the initiation of mature T-cell functions, and T-lymphocyte lineage activities [6,7]. Natarajan et al. demonstrated that the mutations of E224A and S226A at the TCR-Cβ-FG loop exhibited a 34% reduction in T-cell activation, thereby highlighting the structural and mechanistic insights involved with transmitting antigen recognition prior to initiating immune responses [11]. The spatial relationship between these proteins, therefore, is critical in understanding the importance of the Cβ-FG loop in TCR signalling [12,13,14].

In this study, we examined the binding affinity of specific compounds to the Cβ-FG loop site, using computational modelling to determine whether the high affinity binding compounds can modulate T-cell cytokine production. Our findings suggest a novel mechanism of action in which compounds that are non-antigenic, and do not have receptors on the T-cells, may influence T-cell function.

## 2. Results

Molecular dynamics simulation indicated that compounds with a moiety of oestradiol (binding affinity = −6.0 kcal/mol), N-acetyl-D-tryptophan (binding affinity = −5.8 kcal/mol), D,L-Homotryptophan (binding affinity = −6.2 kcal/mol), malonate (binding affinity = −3.1 kcal/mol), and nicotine (binding affinity = −3.4 kcal/mol) had a strong binding affinity to the FG loop of the TCR-Cβ domains. Isobutyrate, 3-indolebutyrate, and propanoate showed no significant modulatory effect on TCR activity, due to either having a weak binding affinity of more than −3.0 kcal/mol at the serine residue, or having no interactions with the FG loop (Appendix A). The compounds oestradiol, N-acetyl-D-tryptophan, D,L-Homotryptophan, malonate, and nicotine demonstrated a strong binding affinity with the TCR-Cβ-FG loop and were thus deemed promising candidates for the hit-to-lead process (Appendix A). Based on the predicted binding affinity, a final ranked list was generated (Table 1, and Appendix A) via in silico screening tools (PyRx, FINDSITE). Ranking of active compounds on specific cytokine assay in order with strongest cytokine production first is shown in Appendix A. Top candidates were identified, including short chain fatty acids (SCFAs, *n* = 44), oestradiol compounds and their derivatives (*n* = 12), and nicotine and its derivatives (*n* = 5).

The results of the in silico docking investigations are summarised in Table 2. All the compounds noted to have a T-cell effect from the biological testing were clustered around the far side of the FG loop on the outer ring. No compounds were found to dock in the middle of the loop. This can be attributed to the presence of a bulky TRP223 and a long ARG227 residue in the middle and arching over the loop, respectively, thereby providing a steric deterrent for ligand binding. All the active ligands were found to interact with residues GLU219, GLU222, TRP223, GLN225, and LYS229, respectively. Most of these interactions were hydrogen bonds, while ligands with aromatic rings such as homotryptophan, oestradiol, and aspartame underwent a few π–π interactions. Ligands with charged groups such as nicotine also exhibited numerous salt bridge interactions. Interactions between these ligands and the FG loop can be seen in Appendix A. Sucrose had the most favourable docking score (−5 kcal/mol), and this may be attributed to the multiple hydrogen bond donors and acceptors present in the structure. In contrast, ligands such as indolebutyrate, isobutyrate, and propanoate, which have been noted to be inactive from biological studies, were found to have docked in different areas. Indolebutyrate and propanoate were bound to the region between the FG loop and the main body of the protein, propanoate was found to bind via a single hydrogen bond to LYS229, while malonate was bound to a region underneath the FG loop. No hydrogen bonds or salt bridges were seen between malonate and the protein; however, it has one of the higher docking scores. This suggests that the cohesive interaction was dominated by dispersive forces. The location of these ligands and interactions are shown in Appendix A.

The top 27 compound candidates (Table 1) with their binding sites identified (Table 2) were investigated in the antigen presentation assay. These compounds, including malonate (*p* = 0.0083) and N-acetyl-D-tryptophan (*p* = 0.0111) were found to exhibit significantly high levels of TGF-β productions when compared to the positive control (Figure 2A). N-acetyl-D-tryptophan (*p* = 0.0324) was found to significantly increase IFN-γ production (Figure 2B). As shown in Figure 2C, the levels of IL-2 were significantly reduced in the presence of the following compounds: N-acetyl-D-tryptophan (*p* = 0.001), D,L-homotryptophan (*p* = 0.0001), oestradiol (*p* = 0.0001), and nicotine (*p* = 0.0103), respectively. Dunnett’s test was the statistic of choice, as it compared all means with the control means to determine whether the differences (between each mean vs. control means) were statistically significant, thereby removing the inter-sample and intra-sample variations.

## 3. Discussion

Initially, the Glide results showed no obvious correlation between the docking scores and biological activity. However, to investigate this further, PrimeMM-GBSA calculations were performed. This combines molecular mechanics (MM) and a generalised Born and surface area (GBSA) solvent mode to calculate the free energy of binding for the ligands. These calculations were performed using the variable dielectric generalised Born (VSGB) solvation model and the OPLS3 force field. A much better correlation was observed between the MM-GBSA binding scores and the biological activity. Based on these calculations, there was a distinct dG cut-off between the active and inactive ligands, at approximately −10 kcal/mol for the best score for an inactive ligand (indolebutyrate, isobutyrate, and propanoate), with the worst scoring active ligand being glycerol at −14 kcal/mol. The sole outlier in this case was indolebutyrate, which, as previously mentioned, binds between the FG loop and the main body of the protein. It has been suggested that its relatively favourable docking score can be attributed to it being larger than the other inactive ligands, thereby resulting in more interactions with the protein.

This study highlights the idea that the regulation of T-cells can be influenced by environmental modulators that can shape the T-cell repertoire by enhancing the production of cytokines, including IL-2, INF-γ, and TGF-β, which are all important for effector and regulatory T-cell [15] functionality, and influences the immune response. Eight out of the thirteen hit-to-lead compounds, including oestradiol, nicotine, tryptophan, aspartame, sucrose, platinum, butanediol, and glycerol exhibited significant modulatory effects in the production of IL-2, IFN-γ, and TGF-β. This is indicative of strong interactions occurring within the TCR-Cβ-FG loop via hydrogen-bond interactions in the active sites at GLU219, GLU222, TRP223, GLN225, and LYS229, respectively (Table 2). In contrast, biologically inactive compounds such as indole-3-butyrate, propanoate, and isobutyrate exhibited strong binding affinities to PHE121, ARG187, ALA228, and GLN233, respectively. It is possible that the interactions of these active compounds with the FG loop at GLU219, GLU222, TRP223, GLN225, and LYS229, respectively, affects the TCR signalling pathway, thereby leading to the modulation of T-lymphocyte function. Therefore, fine-tuning T-cell activation to modulate immunity without suppressing the immune responses is critical to the rationale of future drug design [16,17].

Interest in the interrelationship between environmental factors, health, and disease is increasing. Compounds such as malonate, acetate, and tryptophan are abundant in our diet. It has been shown that malonate and acetate are the primary energy sources for colonocytes, and have anti-carcinogenic, as well as anti-inflammatory properties [18,19]. These latter effects are believed to be mediated through the inhibition of class I histone deacetylases, and the activation of G-coupled receptor targets (GPR43, GPR109A, and GRP41, respectively), which lead to profound anti-inflammatory effects that are necessary to maintain normal colonic homeostasis [20]. Hormones and nicotine have been reported to influence the activation and expansion of T-cells [21,22]. However, the mechanisms that contribute to the immune response have yet to be clearly determined. Here, we show that both oestradiol and nicotine enhance the production of IL-2, leading to an increase in T-cell proliferation. Oestradiol may exert its biological effect through hormone receptors, such as the oestrogen receptor-α complex, to enhance T-cell expansion and differentiation through the interaction of the oestrogen-signalling pathway and the cell surface receptors [23]. Furthermore, the increase in IL-2 production implied an enhancement of TCR-mediating signalling. The interaction between nicotine and TCR may also be involved in T-cell development due to the presence of nicotinic acetylcholine receptors in the circulation and thymus [24].

Although it is undeniable that immune responses are triggered by specific antigens, the present study suggests that there may be an alternative means for nutritional metabolites to regulate T-cell function. This molecular link may impact on both host metabolism and immunity. Das et al. [4] have shown that the FG loop allosterically controls both the variable domain module’s catch bond lifetime and peptide discrimination via force-driven conformational transition. The binding of these drugs at this site may change the strength and the pMHC bond lifetime of antigen recognition, thereby leading to altered signal strength, activation, and subsequent T-cell function. Compounds such as malonate and acetate bind to the FG-loop and interfere with the TCR-β-CD3-ε interface, leading to altered cytokine production. The strength of this effect on the immune response, and whether this interaction can influence a Th1 or Th2 outcome, especially when one considers the prevention or treatment of allergic conditions, needs further evaluation. In this study, the bona fide binding sites targeting the TCR-Cβ-FG loop that modulate TCR engagement in immune signalling provide a rational drug design for the development of next-generation small molecule inhibitors. The TCR-Cβ-FG loop is a potential candidate for the design of future clinical trials in autoimmune diseases including rheumatoid arthritis, psoriasis, and dermatitis. Among the active sites, GLU219, GLU222, TRP223, GLN225, and LYS229 were involved in the interaction between the FG loop and specific compounds (oestradiol, D,L-Homotryptophan, and malonate), which played an important role in transmitting TCR signals. In concordance with the work by Natarajan et al., these residues provided binding activity that modulated T-cell signalling by blocking TCR engagement with MHC on the antigen-presenting cells [11]. The in silico docking-based screening identified bona fide binding sites that facilitated molecular contacts between the TCR-Cβ-FG loop and the environmental cues for transmitting an activation signal in T-cells. Non-antigenic environmental cues such as diet, smoking, and medications can influence T-cell function.

This study is not without its limitations. At present there are no data provided showing that these compounds modulate cytokine secretion by directly binding to the FG-loop of the TCR. It may be that these compounds exert multiple effects on T-cell function unrelated to binding the Cβ-FG loop. For example, nicotine can induce calcium responses by binding to a specific receptor in the T-cells [25]. This may hold true for many of the other compounds. However, there is a need to show that these compounds do indeed bind to the Cβ-FG loop. In future experiments, we will delete the oestradiol receptor from the T-cell surface, and using these modified cells, show that added oestradiol still influences signalling via the FG-loop of the TCR. Despite these weaknesses, the idea of controlling T-cell signalling independently of the antigen by directly targeting the FG loop is interesting and is worth reporting to allow future investigations into understanding the state and activity of T-cells under the influence of the tissue environment and other systemic and local milieu factors.

## 4. Methods and Materials

Seven compound libraries from the respective databases (Appendix A) were screened to determine potential candidates. All libraries were surveyed against the TCR and its isolated Cβ domain using two independent approaches. In the first approach, individual ligands to TCR were geometrically optimised and docked using PyRx [26]. Screening results based on the binding affinities were then ranked, and the top 100 compounds were considered as ‘hits’ in this study. In the second approach, all seven libraries were screened using FINDSITEComb2.0 (CSSB, Georgia Tech, Atlanta, GA, USA, accessed at https://sites.gatech.edu/cssb/FINDSITE-COMB-2/), which predicts ligand-binding pockets based on the binding site similarity among the superimposed groups of template structures identified from threading [27]. The TCR–CD3 receptors occur in 1:1:1:1 stoichiometry for the αβTCR:CD3εγ:CD3δε:CD3ζζ dimers [28]. Structural domain analysis of the proteins was performed using UCSF Chimera and PyMol against the retrieved 3D crystal structure of the TCR (ID: 1OGA, 1.4 Å resolution (Figure 1B) [29]. The docking sites of TCR were then analysed with PyRx and UCSF Chimera, which are widely used computational drug discovery (CDD) software tools for screening the libraries of compounds against potential drug targets. Both software tools used AutoDock to perform the docking of the ligands to a set of grids pre-calculated by AutoGrid as described in Morris et al. [30]. In silico, docking studies were targeted specifically at the FG loop, at residues E218-232 of the TCR-Cβ-FG, based on the biological results which showed reduced IL-2 levels, and previous publications discussing the role of the FG loop in the production of IL-2.

We used an antigen presentation assay to assess the early signalling effects of 13/27 compounds following antigen recognition and subsequent cytokine production [31]. Briefly, we used the 2B4.11 T-cell hybridoma cell line, and the HLA compatible LK35.2 B-cell hybridoma as the antigen-presenting cell (I-Ek bearing). The 2B4.11 T-cell line expresses a complete TCR on the cell surface that specifically recognises and produces cytokines in response to recognition of the pigeon breast muscle cytochrome c (PCC) antigen. The effect of compounds on cell viability is shown in Appendix A. IL-2, IL-6, IFN-γ, TGF-β, and GM-CSF levels in the supernatant were measured using an ELISA kit (Jomar Life Research, Sydney, Australia). The normalised cytokine productions (ng/mL) between the compounds were compared using the ANOVA and Dunnett’s multiple comparison tests (GraphPad Prism 6.0). Statistical analysis of cytokine assay results is shown in Appendix A.

Further descriptions for the preparation of the protein structure, molecular docking, screening of compound libraries, active site validation, target protein validation, ligand preparation, receptor grid generation, docking studies, preparation for cytokine assays, cell viability assay, antigen presentation assay, and statistical analyses are presented in the Appendix A.

## Figures and Tables

**Figure 1 ijms-24-09334-f001:**
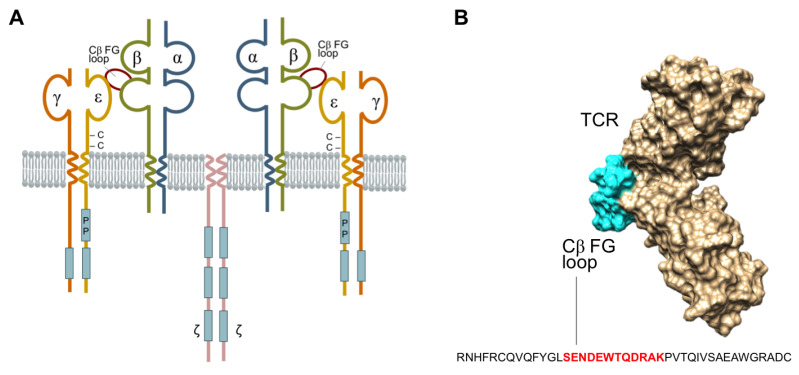
(**A**) The schematic structure of T-cell receptor (TCR) complex, which comprises of T-cell antigen-binding proteins (TCR-α/β), CD3-e,γ,δ chains, and the ζ-ζ homodimer containing the immune-receptor tyrosine-based activation motif’s (blue rectangles) that function in signal transduction. The unique protrusion (highlight in red) is the elongation of the FG loop of the Cβ domain that connects the F and G TCR-β-strands. (**B**) Space filling model of the TCR-αβ heterodimer ectodomain shown in beige (TCR-α and -β chains), according to the Protein Data Bank 1OGA (reference). The Cβ-FG loop is highlighted in cyan, and constitutes an important and integral component that connects the Vβ and Cβ domains. The FG loop of Cα is not visualised in this projection.

**Figure 2 ijms-24-09334-f002:**
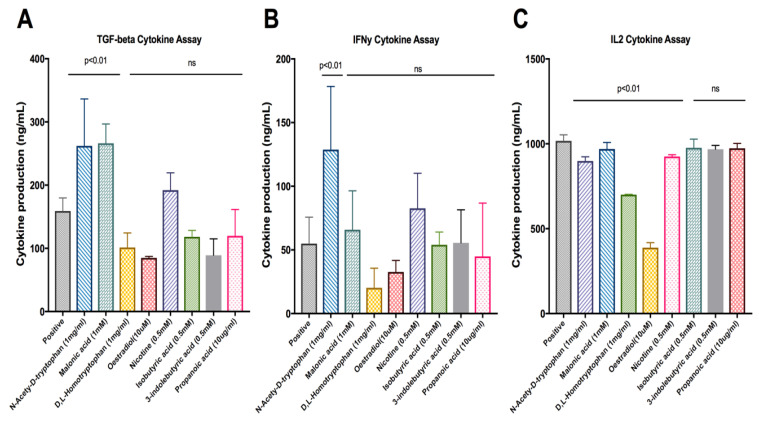
Cytokine production by antigen-stimulated 2B4.11 cells in the presence of the lead compounds. The respective cytokine release, including (**A**) TGF-β, (**B**) IFN-γ and (**C**) IL-2, to the culture supernatant of 2B4.11 cells in the presence of the indicated concentrations of lead compounds. Data represent means ± SD of triplicate measurements. The *p* values were calculated by Dunnett’s multiple comparisons test, and ‘ns’ indicates non-significant.

**Table 1 ijms-24-09334-t001:** List of compounds from in silico screening for TCR-Cβ FG loop binding and their activities on cytokine production.

Rank	Compounds	IL2	INF-Gamma	TGF-Beta
1	Nicotine	Stimulated	No activity	No activity
2	Aspartame	Stimulated	No activity	No activity
3	Estradiol	Stimulated	No activity	No activity
4	Sucrose	Stimulated	No activity	No activity
5	Sucralfate	Not tested	Not tested	Not tested
6	Diethylene glycol monoethyl ether (NF)	Not tested	Not tested	Not tested
7	Ethoxyl-Ethanol	Not tested	Not tested	Not tested
8	3,6,9,12,15-Pentaoxaheptadecane	Not tested	Not tested	Not tested
9	Ethane	Not tested	Not tested	Not tested
10	3,6,9,12,15,18-HEXAOXAICOSANE	Not tested	Not tested	Not tested
11	Carbitol	Not tested	Not tested	Not tested
12	Platinum compounds	Inhibited	No activity	No activity
13	18-Crown-6-tantalum(V)pentachloride	Not tested	Not tested	Not tested
14	2,2′-(Ethane-1,2-diylbis(oxy))diethanol	Not tested	Not tested	Not tested
15	D,L-Homotryptophan	Stimulated	No activity	No activity
16	Malonic	No activity	No activity	Inhibited
17	Isobutyric	No activity	No activity	No activity
18	Propanoic	No activity	No activity	No activity
19	N-Acetyl-D-tryptophan	Stimulated	Inhibited	Inhibited
20	Butanoate	Not tested	Not tested	Not tested
21	Butanediol	Stimulated	No activity	Stimulated
22	Hexanoate	Not tested	Not tested	Not tested
23	Pentanoate	Not tested	Not tested	Not tested
24	Glycerol	Stimulated	No activity	Stimulated
25	Propanediol	Not tested	Not tested	Not tested
26	Indole-3-butanoate	Not tested	Not tested	Not tested
27	Indole-3-butyrate	No activity	No activity	No activity

**Table 2 ijms-24-09334-t002:** Docking scores and key interacting residues for compounds docked into the T- cell receptor (1OGA) via extra precision (XP) docking.

Title	Docking Score ^a^ (kcal/mol)	MMGBSA dG Bind ^b^ (kcal/mol)	Interacting Residues ^c^	Biological Activity
estradiol	−0.4	−31.3	GLU219 TRP223 GLN225	Active
sucrose	−5.0	−29.7	ASP221 GLU222 TRP223	Active
aspartame	−2.1	−27.9	GLU222 TRP223 LYS229	Active
butanediol	−0.9	−21.1	GLU219 GLU222 TRP223	Active
indolebutyrate	−1.6	−20.9	PHE121 ARG187 ALA228	Inactive
nicotine	0.9	−20.6		Active
homotryptophan	−2.0	−18.3	GLU222 TRP223 LYS229	Active
NAD_tryptophan	−1.7	−17.3	GLU124 ARG187	Active
homotryptophan	−0.3	−17.2	GLU222	Active
aspartame	0.4	−16.7	GLU219 TRP223 LYS229	Active
nicotine	−2.4	−15.3	GLU222 LYS229	Active
nicotine	−1.7	−14.4	GLU219	Active
glycerol	−1.7	−13.8	GLN225 LYS229	Active
malonic	0.0	−9.7	GLN225 LYS229	Active
propanoic	−0.4	−5.5	LYS229	Inactive
isobutyrate	−0.8	−4.5	GLN233	Inactive
malonic	−2.0	0.6		Inactive

^a^ Docking scores obtained from extra precision Glide docking are an estimate of protein- ligand binding energies. More negative scores indicate more favourable binding interactions; ^b^ MM-GBSA binding energies are approximate free energies of binding. More negative scores indicate more favourable binding interactions; ^c^ Key interactions residues are taken from the 2D ligand interaction function of Maestro v11.8.

## Data Availability

The datasets generated and/or analysed during the current study are available from the corresponding author on reasonable request.

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
