# Peer review of "Non-Antigenic Modulation of Antigen Receptor (TCR) Cβ-FG Loop Modulates Signalling: Implications of External Factors Influencing T-Cell Responses"

_ijms, 2023, doi:10.3390/ijms24119334_

Round 1

Reviewer 1 Report

1. the binding affinity of specific compounds to the Cβ-FG loop site using computational modelling and determined if the high affinity binding compounds could modulate T-cell cytokine production. What is the rationale of using these different compounds via chemical structure? For example, For example, the
TCR-Cβ-FG loop determined as a receptor/small agonist complex might
not afford the binding of a large antagonist due to the volume shortage of the TCR-Cβ-FG loop.

2. Are there any C-terminal regions of TCR-Cβ-FG loop involving specific structures and perhaps an intermediate loop structure can be important for estimation of compound affinity by the docking calculation?

3. What are the effects of water molecules on TCR-Cβ-FG loop/ligand
docking calculations?. To investigate the effects of water molecules, the docking calculations should be performed using the same TCR-Cβ-FG loops in the presence or absence of water molecules, as they play roles as spacers or mediators of hydrogen bonds between a ligand and residues of the TCR-Cβ-FG loop.

4. The 2B4.11 T-cell line expresses a complete TCR on the cell surface that specifically recognises and produces cytokines in response to recognition of pigeon breast muscle cytochrome c (PCC) antigen. However, HLA compatible LK35.2 B-cell hybridoma used as the antigen presenting cell. Please be precise in the description.

5. Supplemental Table 3 is truncated and requires attention.

6. Title: Non-antigenic modulation of T-cell antigen receptor (TCR) Cβ-FG loop .. gives it more precise rationale of the study.

Must be improved with more precision in writing

Author Response

REVIEWER 1

Comments and Suggestions for Authors

  1. The binding affinity of specific compounds to the Cβ-FG loop site using computational modelling and determined if the high affinity binding compounds could modulate T-cell cytokine production. What is the rationale of using these different compounds via chemical structure? For example, the TCR-Cβ-FG loop determined as a receptor/small agonist complex might not afford the binding of a large antagonist due to the volume shortage of the TCR-Cβ-FG loop.

Response:

The reviewer is correct in suggesting that the TCR-Cβ-FG loop might not afford the binding of a large antagonists due to the volume shortage of the TCR-Cβ-FG loop.  However, we chose this region because of important number of experiments showing this is the site where the proto-oncogene adaptor Nck binds and may have a critical role in influencing TCR signal transduction.  Natarajan et al. demonstrated that the mutation of E224A and S226A at the TCR-Cβ-FG loop exhibited 34% reduction in T-cell activation, highlighting structural and mechanistic insights involved with transmitting antigen recognition prior initiating immune responses. (Natarajan, A. et al. Structural Model of the Extracellular Assembly of the TCR-CD3 Complex. Cell Rep 14, 2833-2845, doi: 10.1016/j.celrep.2016.02.081 (2016)).  Also results from Roy et al unveil a crucial role for the Nck adaptors in shaping the T-cell repertoire to ensure maximal antigenic coverage and optimal T cell excitability. (www.pnas.org/cgi/doi/10.1073/pnas.1009743107).  This is the rational for choosing to study this region.

  1. Are there any C-terminal regions of TCR-Cβ-FG loop involving specific structures and perhaps an intermediate loop structure can be important for estimation of compound affinity by the docking calculation?

We are not aware of any other C-terminal regions of TCR-Cβ-FG loop involved with specific structures important for estimation of compound affinity by the docking calculation.  We have identified an important region within the Cb region, 57 amino acids away from the transmembrane region, important for assembly but is outside the scope of this project.

  1. What are the effects of water molecules on TCR-Cβ-FG loop/ligand docking calculations? To investigate the effects of water molecules, the docking calculations should be performed using the same TCR-Cβ-FG loops in the presence or absence of water molecules, as they play roles as spacers or mediators of hydrogen bonds between a ligand and residues of the TCR-Cβ-FG loop.

We agree with the referee that water molecules can play a role in ligand binding, but not in all cases. In fact, most waters are considered entropically unfavourable when ligand bind. The role of water molecules in ligand docking is already considered in the docking procedure/analysis. No water bridges were identified. A far more detailed picture of water interaction may be found using Molecular Dynamics simulations. But as no water molecules were deemed important, this procedure is beyond the scope of this paper.

  1. The 2B4.11 T-cell line expresses a complete TCR on the cell surface that specifically recognises and produces cytokines in response to recognition of pigeon breast muscle cytochrome c (PCC) antigen. However, HLA compatible LK35.2 B-cell hybridoma used as the antigen presenting cell. Please be precise in the description.

A sentence HLA compatible LK35.2 B-cell hybridoma was used as the antigen presenting cell is placed on page 4 for further clarification.

  1. Supplemental Table 3 is truncated and requires attention.

Thank you.  As part of the formatting process part of the text was truncated.  This has been corrected.

  1. Title: Non-antigenic modulation of T-cell antigen receptor (TCR) Cβ-FG loop .. gives it more precise rationale of the study.

Thank you for the suggestion.  The Title has been changed accordingly.

7. Comments on the Quality of English Language- Must be improved with more precision in writing

As requested, the article was sent to SuperScript Writing & Editing professionals to proof-read, improve the English language, and to conform the article to IJMS guidelines.  Editing notes, corrections and comments are listed below.

IJMS Author guidelines:

  • No full stops after affiliations. Done
  • Postal address format: University of Sydney NSW 2006 Australia (added postcodes).Done
  • + Author to whom correspondence should be addressed (rather than Corresponding Author).Done
  • Use full first and last names for authors, contributors. Done
  • Capitalise all words in headings incl hyphenated, except for conjunctions, articles, prepositions.Done
  • Use em dash for explanations, not commas. Done
  • Numbers: spell out numbers 1–9 unless beginning a sentence. Done
  • Use en dash for ranges (2–4). Done
  • Insert a space between numbers and units. Done
  • Acknowledgements: Dr New – This is Dr Elizabeth New. Prof Ann Kwan. Names and titles correct.   

References

  • IJMS Author Guidelines suggest in-text citations should be in brackets but, as they also say, references in first submissions can be in any style—just consistent throughout—so we left them as is.

General

  • We have made changes to spelling, grammar, punctuation, voice (active), and sentence length for clarity and consistency—please see tracked changes.

Style sheet (UK/AU spelling, hyphenation, etc)

  • affinity score: should this be written as -5 kcal/mol rather than –5 kcal/mol? I have changed them to “-“ throughout
  • analysed
  • antigen-presenting (hyphenated)
  • BAdvSci
  • co-accessory
  • D,L-Homotryptophan
  • data (plural)
  • Dunnett’s test
  • et al.
  • FINDSITE CSSB
  • GraphPad Prism 6.0
  • hit-to-lead process
  • in silico
  • in vitro
  • ligand-binding affinities
  • MM-GBSA
  • oestradiol
  • optimised
  • PyRx
  • signalling
  • software: singular and plural
  • solvation
  • stoichiometry
  • summarised
  • Supplementary Table 1
  • T-cell (hyphenated)
  • UCSF Chimera

Reviewer 2 Report

The authors used in silico computer modeling of T cell antigen receptors to study exogenous factors associated with T cell reactivity.

Research is well done. However, as a result, and even if it is possible to browse on-site, how about all figures and tables being supplements?

It is not enough to simply put all the results in the supplement, but the author must properly judge the importance of them, and then summarize what is necessary. And authors must first construct figures and tables in the text that are visible to the reader.

Author Response

REVIEWER 2 

Research is well done. However, as a result, and even if it is possible to browse on-site, how about all figures and tables being supplements?

 We have kept the figures and Tables at a minimum within the text to allow easier flow of concepts and results.  Putting all the results in supplements would distract the reader, who would have to navigate back and forth. 

It is not enough to simply put all the results in the supplement, but the author must properly judge the importance of them, and then summarize what is necessary. And authors must first construct figures and tables in the text that are visible to the reader.

Round 2

Reviewer 1 Report

authors responded adequately to the reviewers' comments

OK